# Exploring the Biology of Cancer-Associated Fibroblasts in Pancreatic Cancer

**DOI:** 10.3390/cancers14215302

**Published:** 2022-10-28

**Authors:** Adam S. Bryce, Stephan B. Dreyer, Fieke E. M. Froeling, David K. Chang

**Affiliations:** 1Wolfson Wohl Cancer Research Centre, School of Cancer Sciences, University of Glasgow, Switchback Road, Bearsden G61 1BD, UK; 2West of Scotland Pancreatic Unit, Glasgow Royal Infirmary, 84 Castle Street, Glasgow G4 0SF, UK; 3Cancer Research UK Beatson Institute, Switchback Road, Bearsden, Glasgow G61 1BD, UK; 4Beatson West of Scotland Cancer Centre, 1053 Great Western Rd, Glasgow G12 0YN, UK

**Keywords:** pancreatic ductal adenocarcinoma, PDAC, cancer associated fibroblast, tumour microenvironment, immunotherapy, pancreatic cancer

## Abstract

**Simple Summary:**

This review summarises the current understanding of cancer-associated fibroblasts (CAFs) in pancreatic ductal adenocarcinoma (PDAC). The aim of this review is to discuss recent advances in CAF biology, the various roles of CAFs, and different CAF subtypes. We will also examine the evolution of CAFs as tumours progress, and the relationship between CAFs and other cell types within PDAC tumours. Finally, we will provide an update on the potential for therapeutic targeting of certain aspects of CAF biology, and discuss future directions for this exciting and rapidly-progressing field.

**Abstract:**

Pancreatic ductal adenocarcinoma (PDAC) is a lethal malignancy characterised by a stubbornly low 5-year survival which is essentially unchanged in the past 5 decades. Despite recent advances in chemotherapy and surgical outcomes, progress continues to lag behind that of other cancers. The PDAC microenvironment is characterised by a dense, fibrotic stroma of which cancer-associated fibroblasts (CAFs) are key players. CAFs and fibrosis were initially thought to be uniformly tumour-promoting, however this doctrine is now being challenged by a wealth of evidence demonstrating CAF phenotypic and functional heterogeneity. Recent technological advances have allowed for the molecular profiling of the PDAC tumour microenvironment at exceptional detail, and these technologies are being leveraged at pace to improve our understanding of this previously elusive cell population. In this review we discuss CAF heterogeneity and recent developments in CAF biology. We explore the complex relationship between CAFs and other cell types within the PDAC microenvironment. We discuss the potential for therapeutic targeting of CAFs, and we finally provide an overview of future directions for the field and the possibility of improving outcomes for patients with this devastating disease.

## 1. Introduction

Pancreatic ductal adenocarcinoma (PDAC) is a lethal malignancy which remains refractory to treatment despite recent advances in chemotherapy and surgical outcomes. It is projected to become the second leading cause of cancer-related deaths in the United States by 2030 [1] and 5 year survival in the UK remains stubbornly low at around 5–7%, a figure essentially unchanged in the past 5 decades in stark contrast to other cancer types [2,3,4]. PDAC is characterised by a dense desmoplastic and fibrotic reaction with a tumour microenvironment (TME) dominated by an abundant extracellular matrix (ECM) along with heterogenous cell populations including immune cells, macrophages, vascular endothelial cells, and cancer-associated fibroblasts (CAFs) [5]. This fibroinflammatory infiltrate is known as stroma, which can be as much as 80–90% of PDAC tumour volume and plays crucial roles in tumour development, invasion, metastasis, and chemoresistance [6,7].

Of all the heterogenous cell populations present within the stroma, CAFs are arguably the main contributor to aberrant fibrosis and desmoplasia which has led to cancers being classically described as “wounds that do not heal” [8]. This description embodies the decades-old doctrine that CAFs, fibrosis and a wound repair response are uniformly tumour-promoting [9], and a multitude of studies have identified the contribution of CAFs to chemoresistance, immunosuppression and tumour progression [10,11,12]. In recent years there has been an increasing accumulation of evidence suggesting CAF biology is more complex than previously thought and that CAFs may not be uniformly tumour-promoting [13,14]. Furthermore, the advancement of single-cell RNA sequencing technology (RNA-seq) and bioinformatic methodology has allowed the characterisation of CAFs and other cell types within the PDAC TME at greater depths than previously possible [15,16,17]. The purpose of this review is to discuss recent advances in CAF biology and understanding of CAF functional heterogeneity, with an emphasis on the relationship between CAFs and other TME compartments. We also provide an update on markers of CAF subtypes and finally a summary of the potential for therapeutic targeting of CAFs.

## 2. CAFs Have Multiple Cells of Origin and Activating Factors

### 2.1. Origin and Function of Fibroblasts in the Normal Pancreas

First identified by Virchow (1858), fibroblasts are spindle-shaped mesenchymal cells which reside in connective tissues and synthesise ECM and its constituent collagen [18]. They are often defined by these features but also by their lack of mutations found within cancer cells and lack of lineage markers for epithelial cells, endothelial cells or leucocytes [19]. They are key contributors to the structural maintenance of most tissues and when activated they temporarily expand to aid in wound repair and are known as myofibroblasts [20]. Activated fibroblasts can originate from a multitude of cell types (Figure 1) including tissue-resident quiescent fibroblasts, mesenchymal stem cells or pancreatic stellate cells (PSCs), or from trans-differentiation from different stromal cell types such as endothelial cells, adipocytes or pericytes [21,22]. Stellate cells are found in the pancreas, liver, lungs and kidney, and have similarities to fibroblasts although they exhibit different functions such as cytoplasmic storage of vitamin A in lipid droplets [23,24,25]. Their developmental origin remains unclear and they may represent a heterogenous group of different fibroblast cell types, rather than a single cell type [26].

When quiescent, fibroblasts and PSCs synthesise little collagen and form few cell-to-cell connections. Activation is trigged by multiple stimuli including hypoxia and injury-induced TGFβ secretion, resulting in fibroblast proliferation, increased contractility, and expression of fibroblast activation markers such as αSMA, PDGFRα/β, FAP, Vimentin and Desmin [21,27]. Activated PSCs also lose their vitamin A droplets [23,28,29]. Activated fibroblasts deposit ECM proteins such as collagen and fibronectin and secrete factors such as TGFβ, VEGF, CXCL10/12 and IL6 to promote proliferation and recruitment of other cell types to damaged tissue [30]. Of crucial importance is the ability of fibroblasts to become reversibly activated and return to quiescence in normal physiology, an ability lost in cancer and certain fibrotic diseases [31].

### 2.2. CAF Activating Factors

Once chronically activated in the context of cancer, CAFs contribute to the aberrant fibrosis and desmoplasia characteristic of PDAC in a “wound repair” response [8]. A complex and intricate network of signalling pathways and crosstalk with epithelial cells maintains the activation of CAFs. This network includes oncogenic *KRAS*, NFκB, JAK/STAT, PDGF, TGFβ and Wnt [32,33] (Figure 1). *KRAS* is mutated in the vast majority of PDAC and precursor pancreatic intraepithelial neoplasia [34,35]. Mouse models support the notion that oncogenic *KRAS* is a driver of PDAC and of fibroblast activation, and that fibroblast activation is reversed upon inactivation of oncogenic *KRAS* [36,37]. Oncogenic *KRAS* has also been shown to regulate reciprocal signalling networks between the epithelial and CAF compartment of the PDAC TME, rather than being cell-autonomous [38].

NFκB signalling has also been hypothesised to play a role in CAF formation [39,40]. Activated by numerous ligands including TNFα and IL1, the activated NFκB transcription factor is responsible for the regulation of a wide range of cellular and inflammatory processes [41]. IL1 and TNFα have been implicated in tumour fibrosis and CAF formation in PDAC via this pathway [42,43,44]. Aberrant JAK1/STAT3 activation has also been shown to sustain the proinvasive activity of CAFs, and STAT3 has been shown to play a crucial role in *KRAS*- and IL6-driven PDAC development [45,46,47]. TGFβ family ligands drive expression of the myofibroblast marker αSMA in CAFs and subsequent contractility of the cytoskeleton [48]. Other factors shown to activate CAFs include ECM stiffness and composition [49,50] and physiological stress [51,52]. Aberrant desmoplasia and fibroblast activation is also a hallmark feature of other non-malignant fibrotic diseases such as chronic pancreatitis [26].

## 3. CAFs Exhibit Significant Functional and Phenotypic Heterogeneity

The historic view of CAFs as a uniform cell population has recently been challenged by an accumulation of evidence demonstrating CAF heterogeneity in function, transcriptional profile, and spatial relationship with co-existing TME cell types and the tumour epithelium [26]. The existence of different stromal subtypes in PDAC could also imply the existence of different CAF subtypes [53]. CAF heterogeneity in PDAC was first hypothesised within the last 12 years [54,55] however in 2017 the existence of distinct fibroblast populations in PDAC was demonstrated, termed *myofibroblastic CAF* and *inflammatory CAF* [56] (Figure 1).

### 3.1. Myofibroblastic CAFs

Analysis of human PDAC using immunofluorescence identified a subpopulation of FAP positive, αSMA-high cells. Co-culturing mouse PSCs with mouse PDAC epithelial organoids revealed a subpopulation of PSCs which became “activated” (acquired a CAF phenotype with cellular elongation and collagen deposition) when in close proximity to epithelium. These αSMA-high CAFs also demonstrated an altered secretome with low IL6 secretion on analysis of conditioned media from organoids. They were observed in the periglandular region in *Kras^LSL-G12D/+^;Trp53^LSL-R172H/+^;Pdx1-Cre* (KPC) mouse tumours with immunofluorescence and single-cell transcriptome analysis revealed upregulation of TGFβ and *ACTA2* response genes (*CTGF* and *COL1A1*). These CAFs were termed *myofibroblastic* CAFs (myCAFs) [56] (Figure 1).

Genes differentially expressed in myCAFs include *TAGLN*, *MYL9*, *TPM1*, *TPM2*, *MMP11*, *POSTN* and *HOPX*. Gene set enrichment analysis has also revealed the upregulation of smooth muscle contraction, focal adhesion, ECM organisation, and collagen formation in myCAFs in human PDAC [15]. This would be expected given their contractile, stroma remodelling phenotype.

### 3.2. Inflammatory CAFs

In contrast to myCAFs, a subpopulation of αSMA-low CAFs with high IL6 secretion was observed distant from tumour epithelium and from myCAFs [56]. This subpopulation was activated by paracrine signalling from tumour cells as demonstrated in a Transwell system (which allows paracrine interactions between organoids and PSCs but blocks direct contact between the two cell types). Single-cell transcriptome analysis revealed upregulation of cytokines (*IL6*, *IL11*, *LIF*) and chemokines (*CXCL1*, *CXCL2*) in this subpopulation and they were therefore termed inflammatory CAFs (iCAFs) (Figure 1). Genes differentially expressed in iCAFs include *CFD*, *LMNA*, *DPT*, *HAS1*, *HAS2* and *AGTR1*. Gene set enrichment analysis identified upregulation of inflammatory signalling pathways such as IFNγ response, TNF/NFκB, IL2/STAT5, IL6/JAK/STAT3, and the complement pathway in iCAFs in human PDAC [15].

The existence of distinct myCAF-like and iCAF-like subpopulations has been demonstrated in numerous subsequent single-cell RNA-seq studies in mouse and human PDAC [15,57,58,59]. It has also been demonstrated that CAFs are dynamic and their phenotype can change depending on their spatial and biochemical niche within the TME [56,60].

Recent transcriptomic analysis of patient-derived CAF primary cultures identified four CAF subtypes with prognostic impacts [61]. Subtypes B and D overlapped with the myCAF phenotype with expression of *Acta2* and ECM features (with subtype D having the poorest prognosis of the four). Subtype C overlapped loosely with the iCAF phenotype, with expression of inflammatory mediators and complement components, and displayed prolonged survival. Subtype A had features of both myCAF and iCAF phenotypes, possibly reflective of the bulk sequencing employed rather than single-cell.

### 3.3. Antigen-Presenting CAFs

Large subpopulations of CAFs exhibit neither a myCAF nor iCAF phenotype, suggesting the existence of further undefined CAF subtypes [56]. Single-cell RNA-seq of PDAC tissue from KPC mice has recently revealed a third CAF subpopulation expressing MHC class II genes, similar to antigen-presenting cells of the immune system. This subpopulation has been termed antigen-presenting CAFs (apCAFs) [15]. Evidence of these cells has been found in human PDAC with expression of HLA genes that encode MHC class II chains (*HLA-DRA*, *HLA-DPA* and *HLA-DQA*). These cells also express *CD74* (forming the invariant chain of the MHC class II complex), *SLPI*, and *SAA3* (previously implicated as a pro-tumourigenic factor in murine PDAC stroma [62]).

The existence of apCAFs in human PDAC has also been confirmed by co-staining for COL1A1 and CD74 by RNA in-situ hybridisation (ISH), and for PDGFRβ and HLA-DR/-DP/-DQ by immunohistochemistry (IHC). Gene set enrichment analysis has demonstrated upregulation of antigen presentation and processing in apCAFs, along with fatty acid metabolism, MYC targets, and MTOR complex 1 (MTORC1) signalling. It has also been noted that 20.9% of human CAFs co-expressed *HLA-DRA* and *CD74*, which may suggest an ability of CAFs to generate the MHC class II complex [15].

Similar to myCAFs and iCAFs, it was also shown that apCAFs can dynamically interconvert based on altered expression of MHC class II genes in isolated apCAFs compared to apCAFs cultured in 2D. apCAFs were also shown to present antigens to T-cells in a mouse model of T-cell activation [15]. The existence of the apCAF phenotype has also been validated in subsequent single-cell RNA-seq studies in mouse and human PDAC [58,59], however other studies have suggested that apCAFs are not a true CAF subtype, rather they are mesothelial cells from normal pancreas [16,17].

### 3.4. Metabolic CAFs

A novel fourth CAF subpopulation characterised by high expression of PLA2G2A and CRABP2 has also been identified. Marker genes were related to mitochondrial translational elongation and glycolysis, giving rise to the name metabolic CAFs (meCAFs) [57]. The existence of the meCAF has been confirmed with multiplex immunofluorescence with very few overlapped signals observed between myCAF, iCAF and meCAF markers. meCAFs also have more co-staining for the main glycolytic genes *LDHA* and *PKM2*, and unique enrichment of the transcription factor CREB3L1 was observed [57].

### 3.5. Complement-Secreting CAFs

Chen et al. employed single-cell RNA-seq in human PDAC to corroborate the existence of myCAFs, iCAFs and apCAF [63]. Failing to identify the iCAF or apCAF subclusters, they instead named their three subclusters classical CAFs (cCAF, expressing *COL1A1* and *FAP*), complement-secreting CAFs (csCAF, showing high expression of the complement system not dissimilar to iCAFs) and PSCs which were distinct from cCAFs and csCAFs. The existence of csCAFs was demonstrated in human PDAC using RNA ISH and immunofluorescence. Whether this represents a truly distinct CAF subpopulation or an artificial effect of sample heterogeneity, CAF enrichment method or bioinformatic analysis remains to be seen.

### 3.6. CAF Subtype Markers

The many different CAF subtypes identified are summarised with their associated markers in Table 1. In addition to those described above, other markers have also been defined including PDPN and DCN, each of which are expressed in both myCAF and iCAF subtypes as identified by single-cell RNA-seq [15]. The specificity of PDPN as a CAF marker has been validated in IHC with mouse CAFs [15], and PDPN expression in stromal fibroblasts was previously associated with a poor prognosis in human PDAC [64]. CXCL12 has also been identified as a potential iCAF biomarker, being uniquely expressed by cells of the iCAF phenotype [59].

Leucine-rich repeat containing 15 (LRRC15) has also been identified as a marker of myCAF-like cells in mouse PDAC, and TGFβ-driven LRRC15^+^ cells have been shown to dominate the CAF compartment in late-stage tumours [16]. *LRRC15* is a highly differentially expressed gene between CAFs and normal tissue fibroblasts and its presence as a cell surface marker has been confirmed in human PDAC with both IHC and single-cell RNA-seq [16,17]. As expected, LRRC15^+^ cells were found surrounding tumour islets (in keeping with their myCAF phenotype), however they were also frequently seen in proximity to CD8^+^ T-cells.

S100A4, also known as fibroblast-specific protein 1 (FSP1), was first identified as a murine fibroblast marker in 1995 and has since been established as a fibroblast biomarker in breast cancer [65,66,67]. S100A4 expression has subsequently been observed predominantly in PDAC apCAFs [68]. However S100A4 may also highlight tumour cells that have undergone epithelial-mesenchymal transition (EMT), and the overlap between αSMA and S100A4 in stromal fibroblasts is minimal [69,70]. Nonetheless high S100A4 expression has been well validated as a poor prognostic biomarker in PDAC and further elucidation of S100A4-positive CAFs is required [71].

**Table 1 cancers-14-05302-t001:** CAF subtypes with their associated markers and locations.

Subtype	Marker	Name	Location	Ref.
panCAF	FAP	Fibroblast activation protein	Cell surface	[56]
	PDPN	Podoplanin	Cell surface	[15]
	Vim	Vimentin	Cytoskeleton	[15]
	DCN	Decorin	Extracellular matrix	[15]
myCAF	αSMA (*Acta2*)	α-Smooth muscle actin	Cytoskeleton	[56]
	CTGF	Connective tissue growth factor	Extracellular matrix	[56]
	COL1A1	Type 1 collagen (alpha 1)	Extracellular matrix	[56]
	pSMAD	NA	Cytoplasm/nucleus	[72]
	LRRC15	Leucine-rich repeat-containing 15	Cell surface	[16]
	POSTN	Periostin	Extracellular matrix	[57]
iCAF	IL6	Interleukin 6	Secreted cytokine	[56]
	IL11	Interleukin 11	Secreted cytokine	[56]
	LIF	Leukaemia inhibitory factor	Secreted cytokine	[56]
	PDGFR	Platelet-derived growth factor receptor	Cell surface	[56]
	CXCL1	CXC-motif chemokine ligand 1	Secreted cytokine	[56]
	CXCL2	CXC-motif chemokine ligand 2	Secreted cytokine	[56]
	Ly6C	Lymphocyte antigen 6c	Cell surface	[15]
	pSTAT3	Signal transducer and activation of transcription	Cytoplasm / nucleus	[72]
	ApoD	Apolipoprotein D	Cytoplasm	[57]
	CXCL12	CXC-motif chemokine ligand 12	Secreted cytokine	[59]
apCAF	HLA-DR/-DP/-DR	Human leucocyte antigen	Cell surface	[15]
	CD74	NA	Cell surface	[15]
	SLPI	Secretory leucocyte protease inhibitor	Extracellular matrix	[15]
	SAA3	Serum amyloid A3	Secreted apolipoprotein	[15]
	S100A4 (FSP1)	Fibroblast-specific protein 1	Cytoplasm	[68]
meCAF	PLA2G2A	Phospholipase A2	Cell surface	[57]
	CRABP2	Cellular retinoic acid binding protein	Cytoplasm	[57]
Lineage	CD105	Endoglin	Cell surface	[60]

## 4. CAF Functional Diversity: Friend or Foe?

It is becoming clear that CAFs exhibit both tumour-promoting and tumour-suppressive functions, reflective of their heterogeneity and functional diversity. Improving understanding of this diversity will shed crucial light on the factors which restrain and promote tumour growth and will form a basis for targeted therapeutic development. Numerous mechanisms have been proposed to explain this functional heterogeneity. These include: (1) a single class of CAF with multiple functions, (2) several CAF classes each with distinct functions, or (3) several CAF classes with interchangeable functions dependent upon the influence of the surrounding microenvironmental niche [73] (Figure 2).

### 4.1. Tumour-Promoting CAF Functions

Several mechanisms exist by which CAFs can promote tumour development including ECM remodelling, tumour-promoting metabolite secretion, and an immunosuppressive secretome. Implicated CAF-derived factors in tumour development include TGFβ, LIF, CXCL12, IL6, IGF1, and MAPK and STAT3 signalling pathways [27,38,74,75,76,77,78]. TGFβ has also been shown to influence the immune compartment of the TME with effects on cytotoxic T-cell function, neutrophils and macrophages [79,80,81].

CAFs are the main contributors to the fibrotic PDAC stroma which has been implicated in tumour development via regulation of cell adhesion, therapy resistance and the creation of an immunosuppressive TME [12,40,82,83]. High ECM density has also been shown to restrict T-cell access to cancer cells in PDAC [84,85]. In addition, ablation of stromal hyaluronic acid remodels the TME, improving gemcitabine delivery and subsequent survival in murine PDAC [86]. However, despite initial promising results of using a PEGylated recombinant human hyaluronidase as treatment for hyaluronan-high tumours [87], this approach unfortunately failed in a randomised phase 3 clinical trial [88].

Finally, the nutrient-poor, hypovascular and hypoxic PDAC stroma is vulnerable to metabolic manipulation [89,90]. CAFs have been shown to provide metabolic support to cancer cells and immune cells through the production of various metabolites including alanine, proline and lipid species [91,92,93,94,95]. In addition, secretion of the nucleic acid constituent deoxycytidine from PDAC CAFs has been shown to protect from gemcitabine toxicity [96]. CAF-derived exosomes containing diverse metabolites including amino acids and Krebs cycle intermediates have also been shown to influence PDAC cells and promote proliferation, possibly in a paracrine fashion [97,98]. These metabolic mechanisms by which tumour cells are supported by CAFs may be a conserved feature of the wound-healing response employed by normal fibroblasts to support epithelial regeneration [99].

### 4.2. Tumour-Suppressing CAF Functions

Mechanisms of tumour restraint by CAFs are broadly similar to those which promote tumours, however these mechanisms have been shown to be more context-dependent. Such mechanisms include a tumour-restraining stroma, the Sonic hedgehog-Smoothened pathway (Shh-SMO), and immune surveillance.

Numerous studies of CAF or desmoplasia ablation in genetically engineered mouse models of PDAC have yielded unexpected results suggesting a tumour-restraining role of PDAC stroma. Mechanistically this was shown either by inhibition of the Shh-SMO signalling pathway [14,100] or by depletion of αSMA^+^ myofibroblasts in transgenic mice [13]. Here, depletion of these myofibroblasts led to invasive, undifferentiated tumours with enhanced hypoxia and EMT, and diminished survival. Furthermore, suppressed immune surveillance with increased CD4^+^ FOXP3^+^ regulatory T-cells (Tregs) was observed. These findings were recapitulated in human PDAC; immunohistochemical scoring for αSMA^+^ cells from resected PDAC of untreated patients demonstrated an association between low αSMA and shorter survival [13]. In light of recent understanding of CAF heterogeneity, it has since been demonstrated that Shh-SMO signalling is enriched in myCAFs, with a reduction in myCAFs and increase in iCAFs with Shh-SMO inhibition. This correlated with increased immunosuppression, with a decrease in cytotoxic T cells and an expansion in Tregs [101]. These results may, in part, explain the failure of Hedgehog inhibition or stromal ablation in clinical trials [88,102,103,104,105]. Metastatic PDAC has also been shown to exhibit less abundant stroma, and high stromal content has been correlated with a favourable outcome in resected PDAC patients [106].

Furthermore, JAK inhibition has been shown to reduce cancer cell proliferation and tumour growth in vitro by suppressing IL1-induced LIF signalling in iCAFs. This skews iCAFs more towards an ECM-producing myCAF phenotype [72]. However, recent trials of JAK inhibition in patients with metastatic PDAC have not been effective [107], in spite of initial evidence suggesting improved overall survival with JAK inhibition in patients with metastatic PDAC who also have high markers of systemic inflammation [108]. More recently, the tumour-restricting effect of myCAFs has also been demonstrated by deleting type I collagen in myCAFs in a dual-recombinase genetic mouse model of spontaneous PDAC [68]. This resulted in significant reduction of total stromal type I collagen content and accelerated the emergence of pancreatic intraepithelial neoplasia (PanINs) and PDAC, decreasing overall survival. It was suggested that this deletion leads to CXCL5 upregulation in cancer cells via SOX9.

## 5. CAFs Evolve during Tumour Development and Are Differentially Activated

### 5.1. CAF Evolutionary Models

Numerous competing mechanisms for CAF evolution and lineage are proposed, and it remains unclear whether CAFs arise from heterogeneity within normal pancreatic fibroblasts [22,109], from a single normal pancreatic fibroblast population which subsequently differentiates during tumour development [16,59], or from entirely different precursor cell types [16,17] (Figure 3).

The rarity of myCAFs in low grade intraductal papillary mucinous neoplasm (IPMN) has been demonstrated. This is despite myCAFs being highly represented in high grade IPMN, suggesting that fibroblast activation can occur in dysplastic non-invasive lesions [59]. Similarly, recent spatial transcriptomic evidence has demonstrated the PanIN fibroblast population is composed of the same CAF subtypes detected in invasive PDAC [109]. Conversely, other evidence suggests the absence of CAF differentiation in early tumours, with the iCAF phenotype only being observed in established PDAC [59].

To test the hypothesis that TME changes during tumour progression affect fibroblast evolution, single-cell RNA-seq on normal tissue, non-malignant adjacent tissue, and early and advanced tumours from *Pdx1^Cre/+^;LSL-Kras^G12D/+^;p16/p19^flox/flox^* (KPP) mice has recently been performed to develop an evolutionary model [16]. This has demonstrated that pre-existing fibroblast heterogeneity in normal pancreas dictates the development trajectories of murine CAFs. Two separate fibroblast lineages which coevolve during tumour progression were identified, one more primed to provide structural support (myCAF phenotype) and another appearing more immunoregulatory (iCAF phenotype), as confirmed with bioinformatic pseudotime analysis with Slingshot [110].

Of note, this evolutionary model was not validated when compared with several human cohorts, with baseline heterogeneity not being observed in normal tissue fibroblasts in human pancreas [16]. Rather, these normal tissue fibroblasts showed a transcriptional profile that combined the mouse normal tissue fibroblast signatures. Human normal tissue fibroblasts subsequently evolved to a single CAF in early tumours which then gave rise to the myCAF or iCAF phenotype in late tumours. In both mouse and human tissue each subtype was also demonstrated to be under the influence of differential activating factors, with enrichment of NFκB binding sites in iCAFs vs SMAD3 binding sites in myCAFs, and IL1/TNFα signalling driving iCAF differentiation vs TGFβ signalling driving myCAF differentiation.

CD105 (part of the TGFβ receptor complex) has recently been shown to define two functionally distinct pancreatic CAF lineages in KPC tumours, each with different effects on tumour growth [60]. In contrast to other markers which displayed graded expression in CAF clusters (representing a spectrum of phenotypic states), CD105 clearly separated two distinct CAF populations. The existence of CD105^pos^ and CD105^neg^ CAFs has been validated in other datasets [15] and confirmed in human PDAC using IHC [60].

Interestingly, CD105^pos^ and CD105^neg^ subpopulations of normal pancreatic fibroblasts were also isolated which retained their CD105 status even after co-culture with a tumour cell-conditioned medium or with fibroblast-modulating signals (TGFβ1, Il1a and IFNγ). Interestingly, apCAF markers were almost exclusive to the CD105^neg^ subpopulation [60]. CD105 is therefore able to distinguish two truly distinct fibroblast lineages and is present in both normal pancreas and PDAC, supporting the theory of CAF subpopulations arising from heterogeneity within normal pancreatic fibroblasts (Figure 3).

### 5.2. Drivers of CAF Differentiation

It has been demonstrated that myCAFs and iCAFs are differentially activated by various molecular drivers and tumour-secreted ligands [16,72]. Evidence suggests IL1 signalling is the main pathway responsible for iCAF induction in PSCs via the NFκB pathway. IL1 also induces LIF secretion in PSCs which further upregulates the iCAF phenotype in an autocrine fashion via the JAK/STAT3 pathway [72]. Conversely, TGFβ cytokine signalling has been shown to shift PSCs and iCAFs towards a more myCAF phenotype via SMAD2/3 binding at the *IL1R1* promoter (determined by ChIP-seq analysis), which results in antagonism of associated JAK/STAT3 signalling. Similarly, JAK inhibition has been shown to shift iCAFs towards a more myCAF phenotype [72].

These differential activating pathways have been confirmed by VIPER analysis (Virtual Inference of Protein-activity by Enriched Regulon analysis), which infers the activity of protein regulators of gene expression using the expression of their target genes. As expected, IL1R1 and STAT3 are differentially activated in iCAFs, and TGFβ1 and SMAD2 are differentially activated in myCAFs [15]. Other regulators active in iCAFs include HIF1α and NRF2, suggesting a role for iCAFs in oxidative stress relief. In myCAFs, regulators known to promote a mesenchymal cell state are active (TWIST1, ZEB1, SNAI1 and SOX4) [15].

High activation of MHCII-related genes in apCAFs has been demonstrated by VIPER analysis of murine CAFs, in addition to high activation of other regulators of immune activity such as BCAM and F11R (members of the immunoglobulin superfamily) and IRF5 (an interferon-regulating protein) [15]. As with iCAFs, antioxidant response genes *NRF2* and *NRF3* are also active in apCAFs. In human CAFs, master regulators belonging to the antigen presentation machinery (CD74 and XBP1) are differentially active in iCAFs, supporting the presence of “apCAF traits” in humans [15].

Ablation of oncogenic *KRAS* has also recently been shown to profoundly affect the PDAC TME, specifically CAF heterogeneity. Mutant *KRAS* loss leads to decreased TGFβ1 expression from the epithelial compartment and increased IL1a levels. In keeping with existing literature, this consequently correlates with a reduction in myCAFs and an expansion in iCAFs [111]. Similarly, mutant *KRAS* has also recently been shown to contribute to the existence of transitory CAF “sub-states”, representing CAFs in transition between previously validated subtypes. These CAF sub-states demonstrate their own unique markers, enriched pathways, and effects on stromal organisation [112].

The transcription factor ATF4 has also recently been established as a key driver of CAF functionality. Fibroblasts in ATF4-deficient mice displayed defects in collagen deposition and reduced ability to support angiogenesis, resulting in pronounced growth delay of pancreatic tumours [113].

## 6. Tumour-Stroma Crosstalk Influences the TME and Clinical Phenotype

### 6.1. Crosstalk between CAFs, Immune Cells and Epithelium

To determine inter-tumoural heterogeneity, Wang et al. defined different extents of desmoplasia in human PDAC on H&E (dense-type, high desmoplasia; loose-type, low desmoplasia) [57]. They revealed the dominance of specific subclusters of epithelial cells in dense vs loose type stroma, with higher expression of genes involved in glycolysis, amino acid synthesis and ECM binding in dense-type stroma. In contrast, higher expression of genes enriched in oxidative phosphorylation and ECM disassembly were noted in loose-type stroma, in addition to genes involved in immune cell trafficking and antigen presentation. This is consistent with the observation of a higher proportion of immune cells in loose-type PDAC. The meCAF was the dominant subtype in loose-type PDAC, whereas the myCAF was the dominant subtype in dense-type PDAC [57].

CellChat analysis [114] of the Wang dataset also revealed the influence of CAF subtypes on the different proportions of immune cells among PDAC tumours. It would therefore appear that the crosstalk between the epithelial compartment and specific stroma compartments (CAF and immune cell) determines the distinct microenvironmental features of the stroma, including immune cell infiltrate and CAF infiltrate, with a predominance of immune cells and meCAFs in loose type stroma. The importance of crosstalk between epithelium, CAF and macrophage compartments has been re-emphasised by recent findings of overlap between CAF ligands which affect tumour cells and those expressed by macrophages also. This implies the role of macrophage coupling with fibroblasts, and the possibility of therapeutic targeting of this partnership [115] (Figure 4).

In human PDAC, Sun et al. demonstrate that activation of CXCR2 by CXCL3 induces a myCAF phenotype in fibroblasts with upregulation of αSMA by CXCL3-CXCR2 signalling [116]. It was also shown that IL33-stimulated tumour-associated macrophages (TAMs) were the primary source of CXCL3, and that CXCL3 and IL33 were correlated with poor survival. IL33 was therefore shown to be the mediator for this TAM-CAF interaction.

Heterocellular Oncostatin M (OSM)—Oncostatin M Receptor (OSMR) signalling has also been shown to reprogram fibroblasts and regulate tumour growth and metastasis. This is mediated by crosstalk with the immune compartment of the TME, with macrophage-secreted OSM stimulating inflammatory gene expression in CAFs, consequently producing a pro-tumourigenic environment [117]. Similarly, deletion of hypoxia-inducible factor 2 (HIF2) in murine PDAC CAFs resulted in decreased tumour growth and improved survival, possibly through interfering with immunosuppressive CAF-macrophage crosstalk [118].

CD105^pos^ and CD105^neg^ CAF subpopulations have also been shown to display distinct coordinated relationships with other mesenchymal cell types within the TME, possibly due to differential response to regulatory signals between each subpopulation. Similarly, CD105^pos^ and CD105^neg^ subpopulations display opposing relationships with several immune cell subpopulations, suggestive of contrasting immune-modulatory effects [60].

Several differentially engaged upstream regulators and pathways have also been identified between CD105^pos^ and CD105^neg^ subpopulations, with CD105^pos^ CAFs being enriched for TGFβ signalling and CD105^neg^ CAFs being enriched for lymphotoxin beta receptor, TNFα, NFκB, IL6, JAK2, and STING1 signalling. Differential expression of genes encoding for secreted factors has also been observed (*POSTN*, *CXCL14*, *IGFBP5* in CD105^pos^ CAFs; *CXCL2*, *GAS1*, *BMP2*, *NOS2* in CD105^neg^ CAFs). Overall this highlights the potential for CD105^pos^ and CD105^neg^ subpopulations to differentially respond to and modify the inflammatory TME [60].

**Figure 4 cancers-14-05302-f004:**
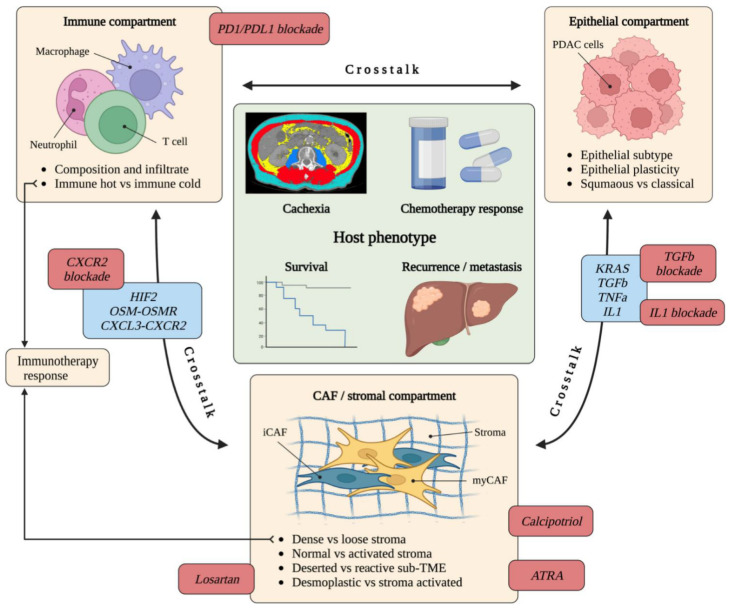
Tumour-stroma crosstalk and CAF therapeutic targets. Crosstalk between the epithelial compartment, immune compartment and CAF compartment is shown along with the specific features of each compartment which are influenced by this crosstalk (yellow boxes). Relevant signalling networks (blue boxes) and potential therapeutic strategies (red boxes, discussed further in Section 7) are also shown. Importantly, this crosstalk influences the host phenotype (green box) [53,57,119,120].

### 6.2. The Relationship between CAFs, Stromal Subtype and Epithelial Subtype

Testing the hypothesis that recurrent PDAC phenotypes are rooted in stromal “functional units”, Grunwald et al. used H&E in human PDAC to define different regions within the TME. “Deserted” regions exhibited thin, spindle-shaped fibroblasts and loose matured fibres. “Reactive” regions exhibited few acellular components, plump fibroblasts with enlarged nuclei, and were rich in inflammatory infiltrate [119]. This correlated with survival, with patients exhibiting multiple regions (i.e., a phenotypically heterogenous TME) having a poorer prognosis than those exhibiting one main region. After extracting CAFs for single-cell and functional characterisation it was revealed that “sub-TME” CAFs were phenotypically and behaviourally distinct, with reactive CAF cultures being more motile and deserted CAF cultures growing faster. Single-cell RNA-seq revealed no association between sub-TME CAF populations and previously defined iCAF and myCAF signatures. Sub-TMEs were therefore concluded to represent organisational units of multi-subpopulation CAF communities [119].

Recent work on genomic and transcriptomic profiling of PDAC has defined two broad molecular subtypes, the better prognosis “classical” subtype and poorer prognosis “squamous or basal-like” subtype [121,122,123]. The effect of stromal heterogeneity on prognosis in PDAC is also well-established by the transcriptomic definition of the better-prognosis “normal stroma” and the worse-prognosis “activated stroma” [53]. Normal stroma is characterised by high expression of αSMA, vimentin and desmin; whereas activated stroma is enriched for genes associated with macrophages and chemokine ligands, possibly pointing towards an activated inflammatory stromal response.

Similarly, Puleo et al. transcriptomically defined the “stroma active” and “desmoplastic” subtypes, taking into account both epithelial and stromal signals [124]. The stroma active subtype features high αSMA and FAP, whereas the desmoplastic subtype is characterised by a low tumoural component and a massive stromal transcriptomic signal with high expression of stromal structural components. Improved survival has been demonstrated in patients with basal-like tumours who have either a stroma active or desmoplastic signature, confirming the hypothesis that stromal heterogeneity is neither uniformly tumour-promoting or tumour suppressing. Conversely, survival is reduced in the presence of the stroma active signature for tumours with a well-differentiated classical epithelial compartment [124]. This also underlies the importance of crosstalk between epithelial and stromal compartments in determining gene expression in each compartment and subsequent effects on clinical phenotype and survival (Figure 4). Furthermore, the importance of taking both epithelial and stromal compartments in to account when determining transcriptomic subtype is emphasised. The influence of TME signalling on epithelial cell state and transcriptional plasticity has also been demonstrated in recent single-cell profiling of PDAC metastatic biopsies and matched organoid models [125].

Recent transcriptomic analysis of patient-derived CAF primary cultures identified four CAF subtypes with prognostic impacts [61]. The intermediate-prognosis CAF “subtype A” (displaying both myCAF and iCAF features) demonstrates enrichment of the worse-prognosis Moffitt activated stroma signature. Similarly, the poor-prognostic squamous subtype signature was more frequently observed in samples with dominant “subtype A” CAFs, suggesting a possible CAF-mediated interaction between activated stroma and the squamous epithelial subtype, associated with a worse outcome. Similarly, Grunwald et al. identified an enriched squamous gene signature in their immune-hot, “reactive” sub-TMEs with activated CAFs. Of note, transcriptomic analysis of epithelial cells isolated as part of experiments to identify the apCAF subtype revealed expression of the classical gene signature [15]. Conversely, most basal-like genes were not detected, suggesting a possible association between apCAFs and the classical subtype.

## 7. The Role of CAFs in Therapy Response

### 7.1. Immunotherapy

Recent interest in the inhibition of the immune checkpoint proteins programmed death ligand-1 (PDL1) and its receptor programmed cell death protein 1 (PD1) has resulted in the approval of these drugs for the treatment of numerous cancer types [126]. However, trials of these drugs in metastatic PDAC have not been promising, likely due to the complexity and functional heterogeneity of the immune compartment of the PDAC TME, and the contribution of CAFs and fibrosis to immune cell infiltration (resulting in an “immune cold” TME) [127,128,129,130,131] (Figure 4).

Wang et al. demonstrated the prognostic value of desmoplasia, noting recurrence within 6 months post-operative in patients with loose-type PDAC and an association between this and marker genes of meCAFs (*PLA2G2A*, *CRABP2*, *SERPINE2*, *MFAP5*). Furthermore, higher expression of meCAF markers on IHC was associated with poorer overall survival, and meCAFs were positively correlated with vascular invasion. This supports the hypothesis of a meCAF^+^ loose-type PDAC being “immune hot” with greater infiltration of immune cells, and higher invasion and metastatic ability. This contrasts with the “immune cold” dense-type PDAC with decreased immune cell access, reduced metastatic/invasive ability, more hypoxia and poorer blood supply. Accordingly, higher abundance of meCAFs was correlated with a better response to immunotherapy [57].

Similar results demonstrate the LRRC15^+^ CAF signature correlating with a poor response to anti-PDL1 therapy across numerous cancer types [16]. Lack of response to anti-PDL1 therapy has been associated with a TGFβ signature in fibroblasts in urothelial cancer, and concurrent TGFβ blockade facilitated T-cell penetration, provoking anti-tumour immunity and tumour regression [81,132]. These LRRC15^+^, TGFβ-dependent CAFs have a myCAF signature, adding further evidence to the hypothesis of an “immune cold” PDAC with dense stroma impeding immune cell infiltration and driving immunotherapy resistance. This is supported by the finding that depletion of Tregs (a key source of TGFβ) leads to myCAF reprogramming and accelerated neoplastic progression [133]. Similarly, targeting immune suppressive mechanisms such as CSF1R^+^ macrophages (contributing to the squamous PDAC subtype) and CXCR2 signalling sensitises to immune checkpoint blockade [134,135] (Figure 4).

### 7.2. Chemotherapy and Radiotherapy

The contribution of CAFs to chemotherapy response is demonstrated by the effects of stromal remodelling and ablation on gemcitabine delivery to the TME in murine PDAC [86]. This may implicate the myCAF phenotype as a key player in chemotherapy resistance, however conflicting results have shown no response to gemcitabine in myofibroblast-depleted tumours [13]. More recent results have also demonstrated improved efficacy of gemcitabine when IL6 is deleted from αSMA positive CAFs, suggesting the role of a more inflammatory CAF phenotype in chemotherapy resistance [136]. Furthermore, CAF-derived nucleic acid metabolites have also been shown to protect from gemcitabine toxicity [96].

In spite of initial evidence suggesting decreased efficacy of gemcitabine and radiotherapy in tumour cells treated with pancreatic stellate cells in vitro [10], recent clinical evidence has demonstrated a 14.0% increase in 5-year overall survival in patients treated with neoadjuvant chemoradiotherapy when compared with those treated with upfront surgery alone [137]. Although recent efforts using single-nucleus and spatial transcriptomic techniques demonstrate the enrichment of specific cell programs after neoadjuvant treatment in human PDAC [120], the effects of radiotherapy in isolation on the fibroblast population within the PDAC TME require further elucidation.

## 8. Therapeutic Targeting of CAFs Yields Mixed Results

### 8.1. PSC Reprogramming and Stromal Remodelling

A characteristic feature of PSC activation is loss of retinol-containing lipid droplets [28,138]. In addition, PDAC patients exhibit a relative deficiency of the fat-soluble vitamins A and D due to absence of biliary and pancreatic secretions, and these conditions may perpetuate a cycle of PSC activation [138,139]. Despite conflicting results as discussed above, stromal remodelling by the vitamin A analogue all-trans retinoic acid (ATRA) has been shown to improve gemcitabine delivery, alter CD8+ T cell infiltration and modulate paracrine signalling resulting in improved survival [29,140,141]. A phase I study of ATRA combined with gemcitabine/nab-paclitaxel showed promising activity, which is currently being further evaluated in a randomised phase II trial as part of Precision-Panc, the national therapeutic development platform for pancreatic cancer in the UK, with stromal-specific retinoid transport protein expression (FABP5, CRABP2) as putative predictive biomarkers [139,142,143] (Figure 4). Similarly, the vitamin D receptor has been shown to act as a master transcriptional regulator of PSCs which induces stromal remodelling, suggesting vitamin D priming as an adjunct in PDAC therapy [144,145]. This strategy is also being evaluated in clinical trials (NCT Numbers NCT03520790 and NCT04524702).

### 8.2. Targeting Cytokines

Tumour-indued IL6 is implicated in a systemic metabolic stress response and reprogramming of host metabolism, resulting in suppression of anti-tumour immunity in murine PDAC [146]. Similarly, IL6 blockade has also been shown to enhance efficacy of PDL1 inhibition in murine PDAC [147].

The identification of IL1 as the initiating factor in NFκB- and JAK/STAT3-mediated shifting of the CAF population towards a more iCAF phenotype may suggest a role for targeting this pathway [72]. Previous studies have indeed implicated this pathway in tumour fibrosis and CAF formation [42,44], and antagonism of the IL1 receptor has been shown to decrease NFκB activity and reduce tumour burden with or without gemcitabine [148].

The effect of NFκB activity in PSCs on promoting tumour growth by increasing expression of CXCL12 in iCAFs has also been demonstrated [149]. It was noted that this in turn prevents cytotoxic T-cells from infiltrating the tumour and killing cancer cells. Subsequent CXCL12 inhibition slowed tumour growth and increased tumour infiltration by cytotoxic T-cells. Similarly, targeting CXCL12 induced rapid T-cell accumulation among PDAC cancer cells and acted synergistically with anti-PDL1 immunotherapy to reduce tumour burden in murine PDAC [76]. The CXCL12-CXCR4 axis is the current target of ongoing clinical trials (NCT04177810 and NCT02907099).

### 8.3. Other Potential Targetable Pathways

Shh-SMO pathway inhibition with a Smoothened antagonist was shown to impair tumour growth [150], switch the CAF balance from myCAF to iCAF, and alter the immune TME in murine PDAC [101]. However, as discussed above, clinical trials of Shh-SMO pathway inhibition have not had success in human PDAC, either with gemcitabine or gemcitabine and nab-paclitaxel [102,103,104,105]. Losartan has also been shown to reduce the generation of type I collagen by CAFs in murine PDAC and enhance the efficacy of nanoparticle-delivered doxorubicin [151,152].

Targeting of the TGFβ pathway with gemcitabine in patients with unresectable pancreatic cancer has previously resulted in improved overall survival in a phase I/IIb clinical trial [153,154]. However, TGFβ inhibition in combination with anti-PDL1 therapy has demonstrated mixed results [155,156,157]. Trials assessing the efficacy of TGFβ inhibition either in isolation or in combination with chemotherapy or immunotherapy are ongoing (NCT04935359, NCT04624217 and NCT04390763).

The tumour-promoting effect of CD105^pos^ CAFs has also been demonstrated, and therefore the potential for therapeutic targeting of this subpopulation [60]. Mice co-injected with a PDAC tumour cell line and CD105^neg^ fibroblasts demonstrate dramatic restriction of tumour growth and improved survival, in contrast to CD105^pos^. CD105^pos^ CAFs were therefore concluded to be permissive to tumour growth, and CD105^neg^ CAFs highly tumour restrictive, likely due to the ability of CD105^neg^ CAFs to establish a tumour-suppressive inflammatory reaction. Interestingly this effect was not noted in mice deficient in innate and adaptive immune functions, suggesting it entirely depends on functional adaptive immunity. This effect was retained even after disrupting the presentation of MHCII antigen presentation machinery using CRISPR-Cas-9 (the apCAF effect) [60].

Inhibition of focal adhesion kinase (FAK) has been demonstrated to reduce tumour fibrosis and immunosuppressive Treg infiltration in KPC mice, sensitising cancer cells to immune checkpoint blockade [84]. Mechanistically this was demonstrated by reduced fibroblast proliferation in response to FAK deficiency. A phase II clinical trial of the FAK inhibitor defactinib in combination with anti-PD1 therapy is currently underway (NCT03727880).

Meflin, a marker of undifferentiated mesenchymal stem cells, has recently been demonstrated to be expressed by PSCs, and Meflin-positive CAFs have been correlated with favourable outcome in human PDAC [158]. Conversely, Meflin deficiency led to poorly differentiated tumours in KPC mice. Similar to CAF-reprogramming strategies as described with ATRA therapy and vitamin D priming, a phase I/II trial of the synthetic retinoid tamibarotene is currently underway in advanced pancreatic cancer (NCT05064618). It is hypothesised that this drug will improve survival by converting Meflin-negative tumour-promoting CAFs to Meflin-positive tumour-restraining CAFs. [159,160].

## 9. Discussion and Future Directions

PDAC remains a treatment-refractory disease with a challenging and unique desmoplastic microenvironment, of which CAFs are a key component. Our understanding of CAFs in PDAC has seen huge advancement in recent years and the complexity of this stromal subpopulation is slowly being revealed. We have demonstrated that CAFs have complex and diverse functions, and viewing them as a uniformly tumour-promoting or tumour-suppressing population vastly underestimates this complexity.

Despite recent advancements in CAF biology, many key questions remain unanswered. Numerous CAF subtypes have been identified however it remains unclear whether these represent distinct CAF lineages, or different functional states of the same cell population which are interconvertible depending upon the surrounding microenvironmental niche (Figure 2). For example, TGFβ was identified as the main driver of the myCAF phenotype and an antagonist of the iCAF phenotype [72], however the iCAF population in established PDAC has been shown to express a high level of TGFβ receptors, suggesting a possible role for this cytokine in regulating certain functions of iCAFs [161].

Further clarification is also required on the definition of a CAF, although recent efforts have cautioned against misinterpreting cancer cells which have undergone EMT as CAFs, or misinterpreting mesenchymal marker positivity as defining a CAF (for example Vimentin may also be expressed by other mesenchymal lineages such as pericytes or adipocytes) [19]. Indeed, whether apCAFs represent a true CAF subtype or mesothelial cells from normal pancreas is yet to be determined [16,17,120].

Further elucidation of the effects of each CAF subtype upon other cell types within the TME and upon the epithelial compartment is also required (Figure 4). The hypothesis of myCAFs creating a contractile, desmoplastic and immune-cold ECM could mechanistically have tumour-promoting effects (by acting as a physical barrier to drug delivery, impeding access of tumour-suppressive immune cells, or creating a hypoxic environment vulnerable to metabolic manipulation) or tumour-suppressing effects (by restricting tumour motility or limiting EMT). Similarly, by creating an immune-hot TME with loose stroma, iCAFs could also theoretically have tumour-promoting effects (creation of an immunosuppressive secretome) or tumour-suppressing effects (enhanced infiltration of cytotoxic T-cells) [162].

Technological advances, particularly in large scale transcriptomics, have allowed for novel and highly detailed profiling of the PDAC TME. However, spatial information will need to be correlated by leveraging emerging high-plex immunofluorescence and spatial transcriptomic technologies which have seen keen interest and advancement in recent years. This will allow the TME composition and crosstalk between different TME compartments and the epithelial compartment of PDAC to be profiled at novel exceptional detail. This will be of particular importance in validating the functional profiles of various CAF subtypes; so far these are mostly hypothesised based on their transcriptional profiles alone and further spatial technologies or modelling systems (such as genetically engineered mouse models, organoids, cell-lineage tracing models or patient-derived xenografts) can be utilised for functional validation of CAF subpopulations [73]. Ultimately this will pave the way for an improved understanding of the complex relationship between CAF subtype and plasticity, epithelial subtype and plasticity, and other TME compartments.

Improved understanding of this relationship and heterogeneity will also lay foundations for the development of potential therapeutic targets or precision medicine strategies, particularly by relating molecular information to host features including survival, recurrence timing and pattern, response to therapy [120], and cachexia / systemic inflammation. Potential future therapeutic strategies could include targeting CAF subtypes, interventions to leverage CAF plasticity by switching CAFs to a more tumour-suppressive subtype (converting “bad stroma” to “good stroma” [163], possibly as an adjunct to immunotherapy or conventional chemotherapy), or targeting systemic inflammation by altering the CAF secretome [164]. Results of ongoing stromal remodelling trials with agents such as ATRA or paricalcitol are also eagerly awaited.

However, numerous challenges are envisaged in this process. As demonstrated, there is a lack of a universal CAF marker and no uniform consensus on which individual markers best represent the CAF population and individual CAF subtypes. Even well-validated CAF markers are expressed by additional cell types within the PDAC TME, for example PDPN by lymphatic endothelial cells, αSMA by pericytes, and FAP by other mesenchymal cells [99]. Furthermore, tissue source can create experimental variability. For example, human tissue by nature is scarce and phenotypically variable depending on which patient cohorts are used, murine tissue exhibits variability depending on which model is used, and in vitro experiments introduce environmental cues either by tumour-conditioned media, signalling molecules, or dimensionality of the culture conditions. Also technical quality of samples and variability in methods used to isolate the stromal compartment and truly capture the fibroblast population can introduce discrepancies [26]. Finally, single-cell RNA-seq is inherently limited by its inability to reflect consequent protein products due to post-translational modification [165,166], and commonly-used cell clustering algorithms have the potential to detect subclusters even where no functional or biologically meaningful subclusters actually exist [73].

## 10. Conclusions

Overall, PDAC CAF biology has enjoyed a recent surge in interest and understanding. As a result, numerous avenues for further research have been opened and there is exciting potential to leverage novel technologies to profile this previously unknown aspect of PDAC at exceptional detail. Ultimately this will lay foundations for precision oncology initiatives and identification of druggable targets to improve outcomes for patients with this devastating disease.

## Figures and Tables

**Figure 1 cancers-14-05302-f001:**
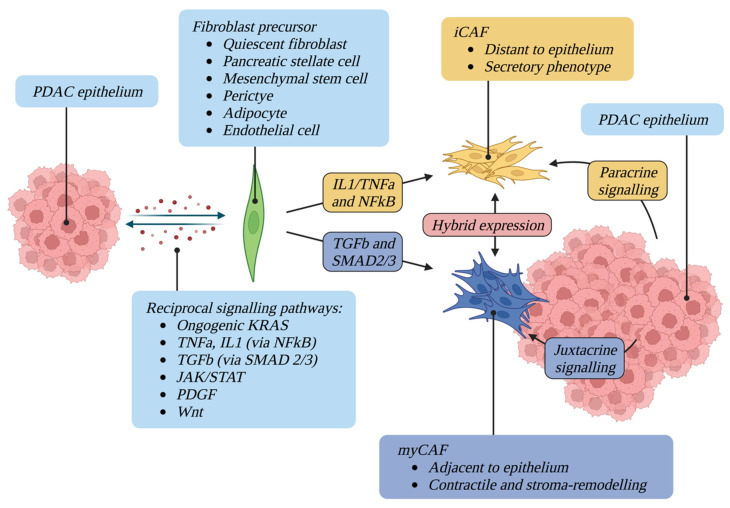
CAF activating factors and plasticity. A complex network of reciprocal signalling pathways maintains the activation of CAFs, which can originate from a multitude of different precursors. Once differentiated, CAFs display different phenotypes and spatial relationships with tumour epithelium. CAFs are phenotypically plastic and can convert between states.

**Figure 2 cancers-14-05302-f002:**
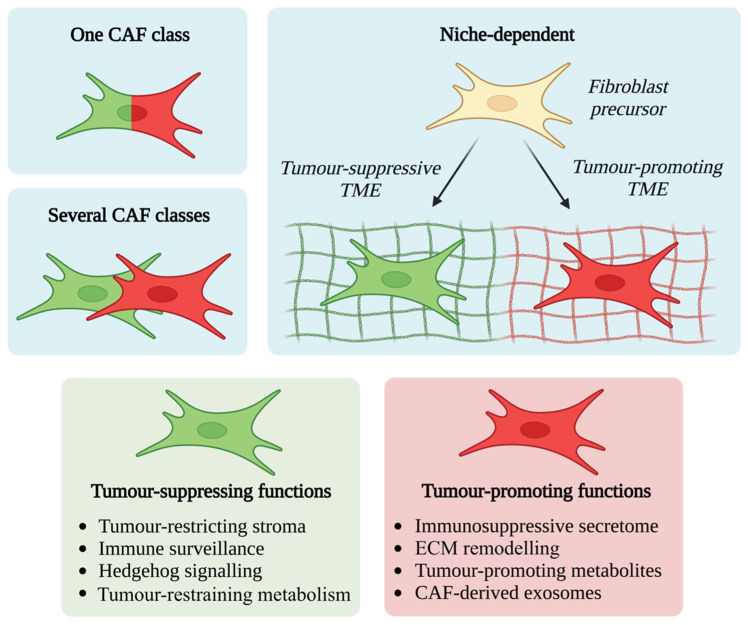
CAF activating factors and plasticity. Numerous mechanisms have been described to explain the functional heterogeneity of CAFs, including (1) a single class of CAF with multiple functions, (2) several CAF classes each with distinct functions, or (3) several CAF classes with interchangeable functions dependent upon the influence of the surrounding microenvironmental niche.

**Figure 3 cancers-14-05302-f003:**
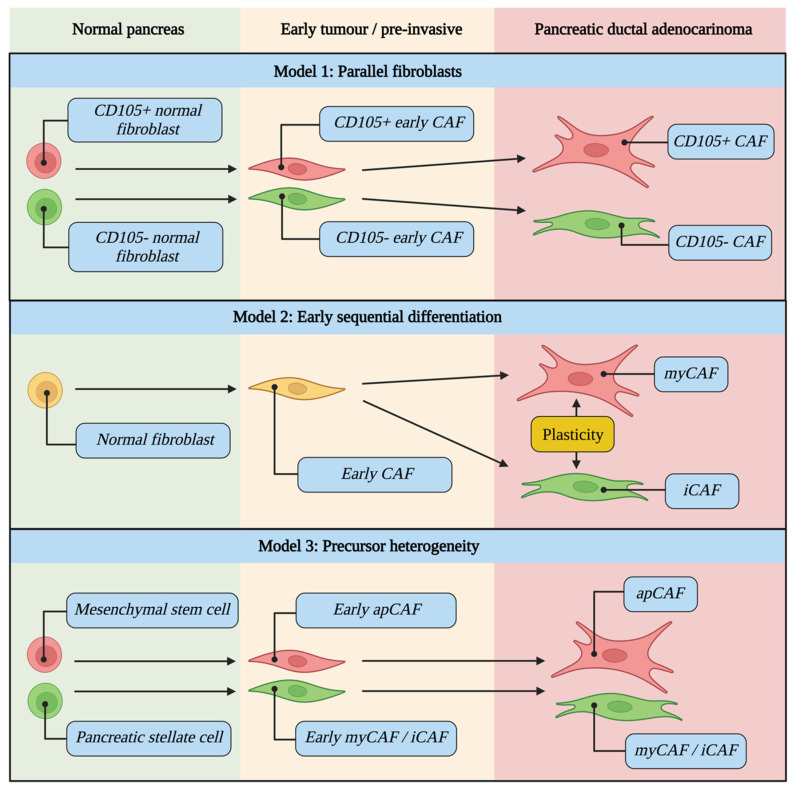
CAF evolutionary models during tumour development from normal pancreas to pre-invasive to PDAC. It remains unclear whether CAFs arise from heterogeneity within normal pancreatic fibroblasts (Model 1), from a single normal pancreatic fibroblast population which subsequently differentiates during tumour development (Model 2), or from entirely different precursor cell types (Model 3). Examples of cell types for each scenario are highlighted in blue.

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
