# Peer review of "Exploring the Biology of Cancer-Associated Fibroblasts in Pancreatic Cancer"

_cancers, 2022, doi:10.3390/cancers14215302_

Round 1
Reviewer 1 Report
A well planned and achieved review, on an ardent problem of health.
Author Response
Thank you for your kind comments.
Reviewer 2 Report
General comments:
The authors present a comprehensive and updated review on the biology of cancer-associated fibroblasts (CAFs) in the context of pancreatic cancer. The manuscript is well written and covers fairly well the main concepts and the latest developments in our understanding on the role of CAFs in pancreatic cancer. Authors make emphasis in describing the recent advances on diversity of CAFs (CAFs subtypes), and in this context, present their potential roles as tumor-promoting or tumor restraining elements of the tumor ecosystem.
The review is nicely drafted and covers the topic rather well, however, the last chapter of the review has room for improvement. Chapter seven of the submitted manuscript (“Therapeutic targeting of CAFs yields mixed results”) should be extended and preferably divided into two chapters: one covering the role of CAFs on responses to therapies (chemotherapy, radiotherapy and immunotherapy) and the other chapter covering the targeting of CAFs/CAF-derived products in combination with other therapies, putting special emphasis on observations from the clinics. For the first suggested chapter, authors have already written a part (subchapter 7.1; in this subchapter authors elaborate on the role of CAFs in the response to immune checkpoint blockade, not referring to CAF targeting strategies). For the second suggested chapter, authors should include other strategies in addition to CAF reprogramming and the target of CAF-derived cytokines and put this in the context of combination therapies. Probably, the most interesting part for the readers would be to know about outcomes from clinical trials.
Author Response
Thank you for your kind comments. Changes have been made as requested.
Reviewer 3 Report
In this manuscript, Bryce et al. reviewed the biology and the therapeutic potential of cancer-associated fibroblasts (CAFs) in pancreatic cancer. Specifically, they highlighted the origin and role of fibroblasts in the normal and malignant pancreas, the heterogeneity of different CAF subtypes, context-dependent pro-tumorigenic and tumor suppressive roles of CAFs, the evolution of CAFs in the course of tumor development and progression, the interaction of CAFs with the tumor and different elements of the tumor microenvironments, as well as the potential therapeutic modalities targeted against CAFs. Overall, the review is well-written and covers an important aspect of the biology of pancreatic cancer, a notoriously deadly malignancy. The authors also effectively used a number of diagrams and tables in the review. Therefore, I recommend the publication of this review.
Author Response
Thank you for your kind comments.
Reviewer 4 Report
This is the summarised review of cancer-associated fibroblasts (CAFs) in pancreatic ductal adenocarcinoma (PDAC). This is discussed about CAFs, and the authors sufficiently discuss the potential for therapeutic targeting of CAFs.
Author Response
Thank you for your kind comments.
Reviewer 5 Report
Congratulations to the authors for their review, very well written and timely important.
I have no criticism, just a typo in FIG 4 reactive and not rective
Author Response
Thank you for your kind comments. The typo has been corrected as requested.